# Assessment of Different Techniques for Adhesive Cementation of All-Ceramic Systems

**DOI:** 10.3390/medicina58081006

**Published:** 2022-07-27

**Authors:** Cristiana Cuzic, Marius Octavian Pricop, Anca Jivanescu, Sorin Ursoniu, Radu Marcel Negru, Mihai Romînu

**Affiliations:** 1Department of Prosthodontics, Faculty of Dental Medicine, University of Medicine and Pharmacy “Victor Babes”, 300041 Timisoara, Romania; pricop.cristiana@umft.ro; 2Research Center of Digital and Advanced Technique for Endodontic, Restorative and Prosthetic Treatment (TADERP), 300041 Timisoara, Romania; 3Department of Oral and Maxillofacial Surgery, Faculty of Dentistry, University of Medicine and Pharmacy “Victor Babes”, 300041 Timisoara, Romania; 4Research Center in Dental Medicine Using Conventional and Alternative Technologies, School of Dental Medicine, University of Medicine and Pharmacy “Victor Babes”, 300070 Timisoara, Romania; radu.negru@upt.ro (R.M.N.); rominu.mihai@umft.ro (M.R.); 5Department of Functional Sciences, Discipline of Public Health, Faculty of Medicine, University of Medicine and Pharmacy “Victor Babes”, 300041 Timisoara, Romania; sursoniu@umft.ro; 6Research Center for Translational Research and Systems Medicine, University of Medicine and Pharmacy “Victor Babes”, 300041 Timisoara, Romania; 7Department of Mechanics and Strength of Materials, Politehnica University, 300222 Timisoara, Romania; 8Department of Prosthesis Technology and Dental Materials, Faculty of Dentistry, University of Medicine and Pharmacy “Victor Babes”, 300041 Timisoara, Romania

**Keywords:** dentistry, all-ceramic restorations, adhesion, cementation protocol

## Abstract

*Background and Objectives*: Modern esthetic dentistry is based on all-ceramic restorations. Dentists still have reservations about using these restorations due to a lack of understanding of the cementation technique, which depends on the type of ceramic used. The aim of the study is to evaluate the approaches and practices of clinicians regarding the adhesive cementation of all-ceramic restorations. *Materials and Methods*: An online questionnaire regarding the use of all-ceramic restorations and their bonding methods was designed by distinguishing the cementation of oxide and silica-based ceramics. The survey included dentists practicing in Timiș County, Romania. The questionnaire and the evaluation of the answers were designed based on the techniques and evidence from the literature. *Results*: Considering the work experience, we obtained two groups: group 1—1 to 6 years and group 2—6 to 9+ years. The results revealed significant values when comparing the two groups in the surface protocol and decontamination (*p* = 0.005), type of cement used (*p* = 0.002), and isolation techniques (*p* = 0.002). *Conclusions*: The results show that many clinicians need additional training to improve their cementing technique and avoid the confusion caused by insufficient information about the interrelationship between the type of ceramic and the cementation procedure.

## 1. Introduction

Due to the evolution of materials and dental treatment requirements, all-ceramic restorations have become a favorable and desired alternative to metal–ceramic fixed restorations. The aesthetic superiority, biocompatibility, and functional advantages of the all-ceramic restorations have expanded their range of clinical indications [1]. Acknowledging the properties of dental materials and their cementation protocol is essential for the long-term clinical success of prosthodontic treatments.

The concept of adhesive dentistry has restructured clinical approaches in terms of preparation, durability, aesthetics, and repair. Ceramic materials are divided by their microstructure and elemental composition into two main classes: mainly silica-based with fillers—typically including feldspathic, leucite, or lithium disilicate—and the oxide-based alumina and zirconia. The infinite variability of the ceramic components can confuse dentists, and without proper guidelines for their use, this can lead to wrong clinical decisions and failure. Knowledge of the classification of ceramic materials, their characteristics, and the content of all modern materials can allow a dental office team to choose the proper ceramic for each particular clinical case [2].

All-ceramic restorations have become increasingly popular as they have proven ideal in situations such as restoring crowns, fixed partial dentures, and implants in aesthetic areas. Zirconia ceramic has replaced metal frameworks for tooth- and implant-supported restorations [3] as its use is adaptable to many treatment options, with a better adhesion quality in prosthodontic restorations.

Even though there are few studies regarding the way dentists apply adhesive cementation techniques, recent studies in the literature reflect improvements when adhesives are used correctly for all-ceramic restorations. To avoid a high clinical failure rate, the prosthodontist should focus his concern on the preparation design and the restoration’s future thickness.

To obtain a high resistance to masticatory forces and be as durable as possible over time, the involved surfaces require systematic decontamination pretreatment and surface conditioning before applying the adhesive cement.

On the other hand, a ceramic restoration should accomplish clinical success over time, dependent on the utilized cementation method and the bond of the restoration with the luting agent [4,5,6,7,8,9,10,11]. Considering the chemical composition of the ceramic, both cementation techniques and surface treatments have become particular [12]. Surface conditioning is required to obtain adhesion between the luting agent, abutment, and ceramic [13]. Restorations from ceramics based on silica are mandatorily etched with hydrofluoric acid and go through silanization. The surface of an oxide-based ceramic—e.g., alumina or zirconia—will not be roughened by etching, as it lacks a silica phase. Even so, Sriamporn et al. [14] proved in their studies that hydrofluoric acid creates micro-morphological changes on the zirconia surface. Zirconia ceramics obtain their micro-mechanical retention and flexural strength through sandblasting surface treatment.

Compared to other ceramic materials, zirconia displays a higher resistance, strength, and mechanical performance—showing a flexural strength of more than 900 MPa and an elastic modulus of 210 GPa [15]. Ceramic primers with phosphoric acid monomers increase the shear bond strength between the resin cement and the ceramic [12].

The prosthodontist should be able to alternate dental materials by knowing their characteristics, and so, depending on each case, choose the optimal restoration material and cementation technique.

All-ceramic dental materials are divided into oxide and silicate ceramics. The crystalline phase of ceramic materials can determine their resistance properties. Oxide ceramics—mostly zirconia ceramics—display the highest values (800–1300 MPa), followed by alumina oxide ceramics (650 MPa). Silicate ceramics such as lithium disilicate show lower flexural strength (300–400 MPa); leucite-reinforced ceramics show around 100 MPa, while the feldspathic ceramic shear strength reaches up to 110 MPa [16].

Due to the various values of resistance and the different characteristics and properties of ceramic materials, choosing the proper cement is as important as applying the correct surface treatment.

The dental abutment must go through several steps before adhesive cementation. Initially, the preparation’s isolation ensures that no surfaces are contaminated with blood, saliva, or temporary residual cement. After rinsing the dental surface, phosphoric acid applied to the enamel will create micromechanical retention. A primer will increase chemical retention.

This study aims to evaluate, through an online questionnaire, the way dentists work concerning ideal conditioning techniques for the ceramic surface—reviewing the literature and manufacturers’ instructions—leading to continuous professional learning and development and increasing clinical performance.

The null hypothesis in the statistical analysis was that the cementation protocol of all-ceramic restorations has no conclusive influence on the clinical success of the prosthetic treatment.

## 2. Materials and Methods

### 2.1. Study Design

An anonymous questionnaire was designed to evaluate dentists in Timiș County, Romania, on how to use all-ceramic cementation protocols, adhesives, and the surface conditioning of prosthodontic restorations. The targeted participants of the study were dentists selected based on their years of experience: group 1 = 1–6 years and group 2 = 6–9+ years. Therefore, the exclusion criteria were practitioners with less than one year of working experience and dental students.

The survey was divided into two parts: a protocol for cementing oxide ceramics (zirconia) and a protocol for cementing silicate ceramics (feldspar, leucite, lithium disilicate).

### 2.2. Questionnaire

The questionnaire was shared online via the Google Forms platform for one year between May 2021 and April 2022, targeting three hundred dentists with different work experience levels. One hundred one clinicians (33%) completed the questionnaire.

Each protocol contained seven multiple-choice questions for each stage of cementing ceramic restorations to understand the workflow of practitioners, such as the type of ceramic and cement used, the surface conditioning methods, and the isolation technique during the cementation process. The questions reflected the standard methods for luting all-ceramic restorations in dentistry and conventional protocols, where needed or indicated. The questionnaire structure followed the working methodology and errors made in the adhesion process of ceramic restorations.

The organized steps of the questionnaire were structured from the literature and reflected working methods, materials used, and techniques to perform a survey on the cementing of all-ceramic restorations.

### 2.3. Statistical Analysis

The data was organized in a Microsoft Excel database and then statistically analyzed. The Mantel–Haenszel chi-square test or the Fisher exact test were used to compare the cementation protocol of all-ceramic restorations between the two groups with regard to work experience levels. A probability level of *p* < 0.05 was considered to indicate statistical significance.

### 2.4. Ethical Consideration

The data collection took place between May 2021 and April 2022. The questionnaire was approved by the Ethics Committee of the University of Medicine and Pharmacy “Victor Babeș” Timișoara (no 36/2021). Complete anonymity was given when submitting the survey. The questions were developed so that the practitioners’ identities were unobtainable through the results.

## 3. Results

The 101 dentists’ questionnaires were divided into two groups based on their work experience: group 1 = 52 (51.48%) 1–6 years and group 2 = 49 (48.51%) 6–9+ years.

Regarding the use of polycrystalline ceramics, 42.57% of the participants did not mention precisely what type of material they use in their dental office (meaning group 1 = 55.77% and group 2 = 28.57%), while 25.74% (group 1 = 17.31% and group 2 = 34.69%) used zirconia. A total of 17.82% mentioned the use of zirconia covered by ceramic restorations (group 1 = 9% and group 2 = 26.53%), while 13.87% mentioned silicate types of ceramics (*p* = 0.015).

As shown in Table 1, 89.11% of the participants (group 1 = 96.15% and group 2 = 81.63%) cleaned oxide ceramic prosthodontic restorations after contamination with saliva, while 10.89% (group 1 = 3.85% and group 2 = 18.37%) mentioned no cleaning after the try-in procedure (*p* = 0.019). A total of 87.13% (group 1 = 96.51% and group 2 = 77.55%) performed decontamination of silicate ceramic restorations, while 12.87% (group 1 = 3.85% and group 2 = 22.45%) did not use any cleaning method after the try-in (*p* = 0.005). A total of 19.80% of the clinicians (group 1 = 9.62% and group 2 = 30.61%) used airflow air-abrasion after the try-in, while 80.20% (group 1 = 90.38% and group 2 = 69.39%) did not (*p* = 0.008).

Silanization was used by 43.56% (group 1 = 32.69% and group 2 = 55.10%) of the dentists as an oxide ceramic surface conditioning prior to the cementation protocol, while 56.44% (group 1 = 67.31% and group 2 = 44.90%) did not mention silanization.

The use of conventional cement was still applied for oxide ceramics by 39.60% (group 1 = 25% and group 2 = 55.10%), while 60.40% (group 1 = 75% and group 2 = 44.90%) would choose an adhesive cement (*p* = 0.002).

Due to the evaluation of the adhesive cementation, multiple types of cement and brand names of cement were suggested as multiple-choice answers. As listed in Table 2, Panavia V5 Kuraray Dental exhibited the most statistically significant result: 16.83% (group 1 = 9.62% and group 2 = 24.49%) chose this cement for oxide ceramics (*p* = 0.046) and 14.85% (group1 = 7.69% and group 2 = 22.45%) chose it for silicate ceramics (*p* = 0.037).

The use of a rubber dam to avoid contamination of the abutments during the adhesive cementation protocol of silicate ceramics was at all times an option for 33.66% (group 1 = 40.38% and group 2 = 26.53%) of the participants; 43.56% (group 1 = 26.92% and group 2 = 61.22%) chose it occasionally depending on the case, while 22.77% (group 1 = 32.69% and group 2 = 12. 24%) performed cementation without a rubber dam (*p* = 0.002).

## 4. Discussion

This study indicates possible errors during the cementation protocol that are made even by dentists with recent training who have knowledge of modern conditioning protocols. This research reflects the importance of continuous medical education for both more and less-experienced dentists.

Polycrystalline ceramics such as aluminum oxide and zirconium oxide ceramics present a high strength, toughness, and excellent resistance to crack propagation. These materials have no glassy phase, so hydrofluoric acid etching will not improve the surface roughness. Sandblasting uses different sizes (50–110 μm) of aluminum oxide Al_2_O_3_ at 2.5 bars which—in contact with the zirconia surface—enhances mechanical retention and strength by pre-treating the ceramic, creating a rough surface. Still, it may increase the risk of crack propagation [4,17]. The silane coupling agent chemically reacts with the ceramic surface after tribochemical silica coating, which roughens and stimulates zirconia.

Altan et al. [18] found that tribochemical silica coating can be preferable to sandblasting the zirconia material in clinical practice.

Clinicians should consider adhesive luting of zirconia restorations to obtain better retention and marginal or internal fit by reducing micro-leakage. Using self-adhesive MDP-based resin types of cement will increase the surface wettability and create bonds at the ceramic’s surface, improving the fracture resistance [19].

A ceramic primer that contains particular adhesive monomers (MDP) is necessary because conventional silane coupling agents cannot bond to metal-oxide ceramics. This creates a durable bond strength between the luting agent and zirconia. However, a study by Saleh et al. [20] pointed out that no matter what ceramic primer is used, sandblasting will increase shear bond strength compared to hydrofluoric acid etching of the surface.

Zirconia, the opaquest dental ceramic, is ideally luted with a dual- and chemical-cured, self-adhesive resin cement [21]. After all, conventional luting agents such as glass-ionomer types of cement are also indicated for zirconia restorations if the abutment preparation offers adequate retention features and ceramic material thickness.

Nicholson et al. [22] considered the main properties of a conventional restorative glass-ionomer cement to be its compressive strength, microhardness, acid erosion, and fluoride release. Therefore, any cement that does not accomplish these features should not be indicated for long-term prosthodontic restorations. However, adhesive cementation of zirconia material restorations is favored.

The ceramics with the highest esthetic performance are predominantly glass ceramics which contain feldspar. These materials have low strength and resistance compared to other ceramics, so adhesive cementation of the prosthetic restorations is required to reduce the risk of fracture [4]. First, the glass-ceramic surface conditioning requires etching with 5–10% hydrofluoric acid for 60 s, followed by rinsing and air drying. The etching treatment causes partial dissolution of the glassy phase, making the ceramic porous and creating micromechanical retentions, increasing the surface area [23]. The silane coupling agent is applied onto the previously etched ceramic surface for 60 s and air-dried. The silane reacts with the exposed ceramic and with the subsequently applied resin cement [24].

The bonding of glass-ceramic restorations requires total-etch adhesive systems followed by the application of dual-cured or light-cured resin cement, following the manufacturer’s indications. Leucite-based glass-ceramics are similar to feldspar ceramics. As their clinical indication suggests for restorations in areas of low stress, the cementation protocol includes etching, silanization, an adhesive total-etch system, and dual-cured or light-cured resin cement.

Ceramics containing lithium disilicate as a filler have a higher strength and resistance and multiple clinical indications as partial or full crown restorations. Etching with 5% hydrofluoric acid for 20 s is followed by 60 s silanization and air drying. Self-adhesive, self-etch, and dual-cured resin cement can be used for adhesive bonding or resin-modified glass-ionomer cement for conventional cementation [25].

Dimitriadi et al. [26] showed in their study that lithium disilicate ceramic surfaces present higher roughness values if first etched with 5% hydrofluoric acid compared to self-etch silane primer only; additionally, the bond strength values increase when the ceramic primer is applied to the HF-etched surface, in contrast to self-etch silane primers.

The choice of cement is influenced by multiple clinical aspects such as the quality and quantity of the tooth structure, the design of the preparation, and the desired retention or abutment isolation. Resin-modified glass-ionomer cement can be used for cementing high-strength oxide-based ceramics. Silica-based all-ceramic restorations are mostly bonded with resin cement or self-adhesive cement.

Esthetic prosthodontic treatments have influenced modern dentistry, leading to high esthetic and long-term stability expectations. An oral rehabilitation aims for a very natural appearance of the teeth and maintains these visual properties over time. This way, the constant evolution of dental materials has also changed work techniques [27]. To obtain an accord between the surrounding anatomical structures and the aesthetic all-ceramic restoration, the clinician must accomplish a flawless protocol for an accurate and successful long-term result.

Due to the high resistance and requirements for aesthetic restorations of zirconium-based ceramics—especially for posterior teeth—these can be cemented with conventional cement and adhesive resin systems.

Regarding the Ostermann F et al. [28] survey, even twelve years after their initial study, the main issue in bonding techniques is the lack of air abrasion used by clinicians’ on the oxide ceramic surface before adhesive cementation.

On the other hand, for silica-based ceramic restorations (feldspar porcelain, reinforced with leucite), dentists should exclusively use adhesive cement combined with appropriate conditioning and primers.

By following the surface conditioning protocol and the cementation process, dentists can improve the resistance of adhesions, optimize the durability of prosthodontic treatments, and improve the performance of clinical results.

The continuous progress made by manufacturers comes to doctors’ benefit and support by simplifying adhesive systems and making prosthodontic treatment plans a convenient, trustworthy option. Unfortunately, this also involves a financial burden, resulting from the requirement to purchase certain, more expensive adhesives and additional materials to perform accurate cementation. Regardless of the additional costs, dentists should act to patients’ advantage to achieve adhesion and the long-term longevity of treatments.

One of the study limitations consists of a limited number of returned questionnaires; however, this still represents 10% of the practitioners in Timiș County. Further research is needed involving a wide-reaching range of dentists.

The fact that the study included only two classes of materials represents another limitation. Future studies could extend this research by supplementing the questionnaire sheets with other prosthodontic materials such as hybrid ceramics.

## 5. Conclusions

It can be assumed that almost half of the participants did not know what type of ceramic material they use; the more experienced group mainly mentioned the use of zirconia or ceramic fused to zirconia restorations. In addition, no matter the level of working experience, both groups of clinicians mentioned cleaning oxide ceramic restorations after the try-in clinical step. However, the cementation of oxide ceramic restorations with conventional cement is a valuable option for most experienced dentists. Additionally, the group with fewer years of practice would always use a rubber dam during cementation, while the more experienced group would only choose it occasionally.

Within this study’s limitations, the questionnaire results reflect the importance of continuous professional development through additional training, guidance, and control for better understanding of and correct workflows for adhesive systems, surface preparation, and conditioning types of ceramics. It is the responsibility of clinicians to have knowledge of the dental materials used and their office equipment. The long-term success of a prosthodontic restoration reflects a dentist’s experience and skills and updates in current innovations. Further surveying is necessary to analyze the workflow of more clinicians and their training, which could be the best way to prove that continuous medical education is essential in dentistry.

## Figures and Tables

**Table 1 medicina-58-01006-t001:** Ceramic surface protocol prior to the cementation of prosthodontic restorations.

Oxide Ceramics				Silicate Ceramics		
Group 1		Group 2	*p*-Value	Group 1	Group 2	*p*-Value
No surface conditioning	13 (25.00)	9 (18.37)	0.42	41 (78.85)	45 (91.84)	0.06
Etching pretreatment with hydrofluoric acid	19 (36.54)	13 (26.53)	0.28	27 (51.92)	34 (69.39)	0.07
Air-abrasion pretreatment	4 (7.69)	3 (6.12)	0.75	2 (3.85)	1 (2.04)	0.59
Silanization	17 (32.69)	27 (55.10)	0.02	16 (30.77)	17 (34.69)	0.67
Other/combined surface treatments	51 (98.08)	49 (100)	0.32	51 (98.08)	49 (100)	0.32

(*n*%) The numbers in the parentheses represent the percentage of the total answers.

**Table 2 medicina-58-01006-t002:** Types of cement used for ceramic restorations.

Oxide Ceramics				Silicate Ceramics		
Group 1		Group 2	*p*-Value	Group 1	Group 2	*p*-Value
Glass-ionomer cement	13 (25.00)	27 (55.10)	0.002	6 (11.54)	2 (4.08)	0.16
Rely X U200 3M	15 (28.85)	18 (36.73)	0.39	14 (26.92)	18 (36.73)	0.28
Panavia V5 Kuraray Dental	5 (9.62)	12 (24.49)	0.04	4 (7.69)	11 (22.45)	0.03
Panavia F2.0 Kuraray Dental	2 (3.85)	2 (4.08)	0.95	6 (11.54)	2 (4.08)	0.16
Variolink Ivoclar Vivadent	3 (5.77)	7 (14.29)	0.15	3 (5.77)	8 (16.33)	0.08
Multilink Ivoclar Vivadent	7 (13.46)	11 (22.45)	0.23	8 (15.38)	13 (26.53)	0.16
Speedcem Ivoclar Vivadent	3 (5.77)	3 (6.12)	0.94	4 (7.69)	6 (12.24)	0.44
Biscem Bisco Dental	9 (17.31)	8 (16.33)	0.89	10 (19.23)	7 (14.29)	0.5

(*n*%) The numbers in the parentheses represent the percentage of the total answers.

## Data Availability

All data is available upon request.

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
