# Peer review of "Assessment of Different Techniques for Adhesive Cementation of All-Ceramic Systems"

_medicina, 2022, doi:10.3390/medicina58081006_

Round 1

Reviewer 1 Report

The article entitled “Assessment of Different Techniques for Adhesive Cementation of All-ceramic Systems” aimed to the evaluate the approach of dental practitioners regarding the adhesive cementation of all-ceramic prosthodontic restorations. The paper is in line with journal’s aim, moreover, Authors have well revised several issues; however, I ask authors to add aimed to provide some key concepts.  

-       In the introduction section it would also be interesting to discuss the pre-treatment techniques of the dental surfaces before the adhesive cementation of the prosthetic restorations (please, see and discuss 10.3390 / ma13133026), thus indicating which factors can influence the cementation procedures, increasing the retention of the dental crowns

-     Study limitations must be included in the paper

-       Conclusions cannot be reduced to a sentence: you must improve them highlighting the limits and the future insights pointed out from this article.

-       The formatting of the references is not correct, please check the journal instructions for authors

-       Several moderate typos are present in the text, please, amend

According to this Reviewer’s consideration, novelty and quality of the paper, publication of the present manuscript is recommended after minor revision.

Author Response

Submission of revised paper Assessment of Different Techniques for Adhesive Cementation of 
All-ceramic Systems [Manuscript ID: medicina-1822427]

 24 July 2022

Dear Reviewer,

         Thank you for your email dated [Medicina] Manuscript ID: medicina-1822427 - Major Revisions [19 July 2022] enclosing the reviewers’ comments. We have carefully reviewed the comments and have revised the manuscript accordingly. Our responses are given in a point-by-point manner below.

            We hope the revised version is now suitable for publication and look forward to hearing from you in due course.

Sincerely,

Dr. Pricop Marius Octavian

Thank you for reviewing the paper.

  1. In the introduction section it would also be interesting to discuss the pre-treatment techniques of the dental surfaces before the adhesive cementation of the prosthetic restorations (please, see and discuss 10.3390 / ma13133026), thus indicating which factors can influence the cementation procedures, increasing the retention of the dental crowns

The dental abutment must go through several steps before adhesive cementation. Initially, the preparation's isolation ensures that no surfaces are contaminated with blood, saliva, or temporary residual cement. After rinsing the dental surface, phosphoric acid applied to the enamel will create micromechanical retention. A primer will increase chemical retention.

  1. Study limitations must be included in the paper

One of the study limitations consists of a limited number of returned questionnaires; however, it represents 10% of the practitioners of Timiș County. Further research is needed involving a wide-reaching range of dentists. The fact that the study includes only two classes of materials represents another limitation. Future studies could extend the research by supplementing the questionnaire sheets with other prosthodontic materials such as hybrid ceramics.

  1. Conclusions cannot be reduced to a sentence: you must improve them highlighting the limits and the future insights pointed out from this article.

It was modified in the Word document.

  1. The formatting of the references is not correct, please check the journal instructions for authors

It was modified in the Word document.

  1. Several moderate typos are present in the text, please, amend

It was modified in the Word document.

Reviewer 2 Report

The manuscript seems interesting and genuine, however the authors should address the following points to improve it:

- The abstract should be non-structured with specific word limitation (see authors' guidelines)

- The research question and current gap in literature should be accentuated in the introduction section

- It is advisable to add null hypothesis at the end of the introduction section

- How was the questionnaire structured and how was it validated? did authors perform a pilot study?

- How many participants were contacted and how many of them responded? (response rate)

- What were the starting and finishing dates of questionnaire collection?

- The author should describe the questionnaire sections briefly 

- Who were the targeted participants and what were the inclusion and exclusion criteria?

- The materials and methods section needs significant changes following the format of survey-based published articles

- The discussion part include so may short paragraphs that should be linked and blended together to shorten the number of paragraphs

- The authors should add the significance of this study in the discussion part

- Study limitations and future recommendations should added to the discussion part

- In the line 266: what do authors mean by "try-in probe"?

Author Response

Submission of revised paper Assessment of Different Techniques for Adhesive Cementation of 
All-ceramic Systems [Manuscript ID: medicina-1822427]

24 July 2022

Dear Reviewer,

            Thank you for your email dated [Medicina] Manuscript ID: medicina-1822427 - Major Revisions [19 July 2022] enclosing the reviewers’ comments. We have carefully reviewed the comments and have revised the manuscript accordingly. Our responses are given in a point-by-point manner below.

            We hope the revised version is now suitable for publication and look forward to hearing from you in due course.

Sincerely,

Dr. Pricop Marius Octavian

Thank you for reviewing our paper. 

  1. The abstract should be non-structured with specific word limitation (see authors' guidelines)

It was modified in the Word document.

  1. The research question and current gap in literature should be accentuated in the introduction section

Acknowledging the properties of the dental materials and their cementation protocol is essential for the long-term clinical success of prosthodontic treatments. Even though there are few studies regarding the way dentists apply the adhesive cementation techniques, the recent studies in the literature reflect the improvement when adhesives are used correctly for all-ceramic restorations.

  1. It is advisable to add null hypothesis at the end of the introduction section

The null hypothesis in the statistical analysis was that the cementation protocol of all-ceramic restorations has no conclusive influence on the clinical success of the prosthetic treatment.

  1. How was the questionnaire structured and how was it validated? did authors perform a pilot study?

The questionnaire is structured in accordance with the author’s interest regarding aspects of the cementation techniques and considering former literature studies. No pilot study was performed.

  1. How many participants were contacted and how many of them responded? (response rate)

The questionnaire targeted three hundred dentists with different work experience levels. One hundred one clinicians (33%) completed the questionnaire.

  1. What were the starting and finishing dates of questionnaire collection?

The questionnaire results were collected between May 2021 and April 2022.

  1. The author should describe the questionnaire sections briefly 

Each protocol contains seven multiple-choice questions for each stage of cementing ceramic restorations to understand the workflow of practitioners such as the type of ceramic and cement used, the surface conditioning methods, and the isolation technique during the cementation process.

  1. Who were the targeted participants and what were the inclusion and exclusion criteria?

The targeted participants of the study are dentists selected by their years of experience. Therefore, the exclusion criteria are practitioners with less than one year of working experience and dental students.

  1. The materials and methods section needs significant changes following the format of survey-based published articles

Modified in Word document

  1. The discussion part include so may short paragraphs that should be linked and blended together to shorten the number of paragraphs

Modified in the Word document

  1. The authors should add the significance of this study in the discussion part

The study indicates the possible errors during the cementation protocol even made by dentists with recent training who have the knowledge of modern conditioning protocols. The research reflects the importance of continuous medical education for both more and less experimented dentists.

  1. Study limitations and future recommendations should added to the discussion part

One of the study limitations consists of a limited number of returned questionnaires; however, it represents 10% of the practitioners of Timiș County. Further research is needed involving a wide-reaching range of dentists. The fact that the study includes only two classes of materials represents another limitation. Future studies could extend the research by supplementing the questionnaire sheets with other prosthodontic materials such as hybrid ceramics.

  1. In the line 266: what do authors mean by "try-in probe"?

Within the try-in probe, the authors meant the clinical step, which includes the temporary placement of a dental restoration or device to determine its fit and comfortableness in the patient’s mouth.

Round 2

Reviewer 2 Report

The authors have answered all comments

and the paper can be accepted in the current

form.